# A Comparative Study of Ensemble Models for Predicting Road Traffic Congestion

Tebogo Bokaba [1], Wesley Doorsamy [2,*] and Babu Sena Paul [2]

1   Department of Applied Information Systems, University of Johannesburg, Johannesburg 2006, South Africa; tbokaba@uj.ac.za
2   Institute for Intelligent Systems, University of Johannesburg, Johannesburg 2006, South Africa; bspaul@uj.ac.za
*   Correspondence: wdoorsamy@uj.ac.za

**Abstract:** Increased road traffic congestion is due to different factors, such as population and economic growth, in different cities globally. On the other hand, many households afford personal vehicles, contributing to the high volume of cars. The primary purpose of this study is to perform a comparative analysis of ensemble methods using road traffic congestion data. Ensemble methods are capable of enhancing the performance of weak classifiers. The comparative analysis was conducted using a real-world dataset and bagging, boosting, stacking and random forest ensemble models to compare the predictive performance of the methods. The ensemble prediction models are developed to predict road traffic congestion. The models are evaluated using the following performance metrics: accuracy, precision, recall, f1-score, and the misclassification cost viewed as a penalty for errors incurred during the classification process. The combination of AdaBoost with decision trees exhibited the best performance in terms of all performance metrics. Additionally, the results showed that the variables that included travel time, traffic volume, and average speed helped predict vehicle traffic flow on the roads. Thus, the model was developed to benefit transport planners, researchers, and transport stakeholders to allocate resources accordingly. Furthermore, adopting this model would benefit commuters and businesses in tandem with other interventions proffered by the transport authorities.

**Keywords:** ensemble methods; misclassification cost; machine learning; road traffic congestion

## 1. Introduction

Road traffic congestions (RTCs) are significant issues globally; they negatively affect economic production and quality of life in different cities. RTCs in transportation services occur when the required demand goes beyond the design capacity. According to [1], road traffic congestions gradually increase and cost economies billions of Rands (ZAR), with cities such as Bengaluru (India) leading globally, followed by Manila (Philippines), Bogota (Colombia), Mumbai (India), and Pune (India), the top-five ranked congested cities globally with over 800,000 in population. Furthermore, the top-five ranked congested cities in Africa are Cairo (Egypt) taking the lead, followed by Cape Town (South Africa), Johannesburg (South Africa), Pretoria (South Africa), and East London (South Africa) for overall daily congestion. In [2], it was also reported that RTCs have impacts on reducing road throughput, increased vehicle emissions, and accidents that cost road users time and money. Commuters residing in large metropolitan areas are mainly affected by RTCs daily disrupting their day-to-day activities.

The increasing traffic congestion is, directly and indirectly, the cause of a significant part of road traffic collisions that result in an increased number of injuries and fatalities on the roads globally. The World Health Organisation (WHO) [2] has also reported that RTCs contribute to health complications, affecting about 3.7 million lives lost globally, specifically in developing cities with high monetary losses, delays, fuel waste, road collisions, and

emission. They also produce nitrogen oxides (NOx), carbon oxides (CO), sulphur oxides (SOx), and particle filters. According to [3], the transport sector accounts for three-quarters of global $CO_2$ emissions, adding that the emissions can originate from trucks, cars, planes, and trains. Furthermore, the report by [2] has stated that the main sources of air pollution are from road traffic congestion-related issues, causing diseases such as cardiovascular and respiratory failure. Traffic congestion is harmful to drivers and passengers sitting in traffic and people living near the affected highways. According to [4], adults living next to busy roadways might end up having a disease like dementia, and when it comes to children, it might risk them developing long-term chronic diseases. In addition, other research studies like [5–10] have investigated the negative impacts of RTCs in detail.

Road traffic congestion prediction presented in this paper can be defined as predicting traffic state at a specific time [11]. This study focuses more on traffic flow and road traffic congestion. Furthermore, the study considers traffic volume (density), average speed (speed flow/velocity) and travel time parameters of the fundamental diagram. The fundamental diagram graphs from [12] consist of freeflow ($V_f \sim$ flow velocity), bound flow ($V_c \sim$ critical velocity), and congested (traffic density) vectors. The fundamental analysis of the study is as follows: freeflow refers to the traffic being stable with vehicles moving freely, and bound flow is the bistable transition from a stable traffic state. Then, the congested state refers to the unstable traffic state, which means vehicles velocity is reduced and the volume of vehicles increases gradually for the freeway capacity [12–14]. For this study, the vectors (classes) were defined as freeflow, moderate, and congested, where the bound flow vector is referred to as moderate. These three classes correspond based on the obtained data.

This study considered supervised ensemble methods to design predictive models for road traffic congestion. Ensemble methods are implemented in machine learning to boost several classifiers and to improve their overall performance. The main aim of ensemble methods is to perform best compared to the traditional machine learning methods called base learners [15]. Ensemble methods have shown superior performance in the road traffic congestion domain recently. In the recent study [16], ensemble methods performed best compared to regression models and multi-layer perceptron. Ensemble methods were considered in this study to improve machine learning classifier performance. This study aimed to develop an ensemble learning model using a real-life road traffic flow dataset to evaluate the negative impacts of road traffic congestion, which, in turn, might lead to high numbers of road traffic collisions on the highways and delays, to name a few. The three main objectives of this study, which deals with a real-life road traffic flow dataset, are as follows:

1. To handle missing values using the listwise deletion method;
2. To compare the performance of three traditional machine-learning methods and three ensemble methods;
3. To assess each model's performance using key evaluation metrics and a cost model incorporating cost to commuters, businesses, and the economy.

The study's contribution is to present an RTC framework that inputs missing values and performs comprehensive analysis using ensemble and machine learning methods, computes classification cost, and evaluates the performance of the models. The entire paper is outlined as follows: Section 2 contains the RTC-related literature review, Section 3 describes the study methodology and methods used. Section 4 describes the experimental settings for the study, while Section 5 provides the experimental results and discussion of the findings. Finally, a conclusion is presented in Section 6.

## 2. Literature Review

An evaluation of the recent research studies that deal with road traffic congestion and road traffic collisions prediction and modelling using different machine learning (ML) approaches has been conducted. A brief review is carried out here to highlight the emphasis

on research towards incorporating ML-based methods into road traffic safety, planning, and management solutions.

### 2.1. Traffic Flow Prediction-Related Studies

This section reviews related studies on road traffic congestion prediction using ML and statistical analysis methods. Most of the studies considered using traditional ML classifiers.

A study by [17] proposed that deep learning can predict urban traffic for traffic events mined from Twitter. The study combined traffic Twitter messages with weather information, and it adopted bi-directional long short-term memory and the stacked auto-encoder method. The results proved the value of the proposed approach by refining the model's performance. Another study [18] assessed the effect of a dataset on evaluating traffic prediction approaches. Their study used existing road traffic data, weather data, driving angles, and congestion levels to generate a new dataset, which was used to propose the prediction model. The study used long short-term memory, gated recurrent unit and the stacked auto-encoder methods. Results of the study showed that the new dataset was accurate.

Another study [19] proposed an approach for monitoring traffic congestion using machine learning to exploit hidden road traffic patterns. The authors managed to identify different time slots in which vehicle traffic congestion occurred, and they monitored average speeds. Yet another study conducted by [20] proposed using deep neural networks (DNN), distributed random forest (DRF), gradient boosting machine, and the generalised linear model to predict short-term traffic flow. The study applied machine-learning approaches to evaluate the use of attributes such as traffic flow, speed, occupancy, and time to reduce the prediction error during traffic flow prediction. On the other hand, [21] only considered artificial neural networks (ANNs) to predict traffic flow, which achieved better results.

Another study [22] used decision trees, random forest, support vector machine, multi-layer perceptron, and logistic regression to forecast traffic congestion in Internet-based cities. The study results showed that logistic regression performed best compared to the other methods. Furthermore, research by [23] improved traffic flow prediction by considering weather data and using autorregressive integrated moving average (ARIMA), ANN and dynamic Bayesian network (DBN) methods for a deep-learning approach. Thus, in [23], using both the weather and traffic flow data, the results demonstrated the effectiveness of the proposed method. Study [24] used the k-nearest neighbor method to predict traffic flow using short-term data. The study model results proved that the method could predict short-term traffic flow. Finally, a study by [25] used online-SVR, Gaussian maximum likelihood, ANN, and Holt exponential smoothing to predict short-term traffic flow. The study results found that the proposed method achieved the best results compared to the other methods.

It is noticed that most of the previous studies considered the traditional ML methods to predict road traffic congestion. Previous studies favourably used ANN, deep learning, long short-term memory, and random forest. For example, to assess the performance of these models, commonly used metrics were mean absolute error (MAE), root mean square error (RMSE), mean absolute difference (MAD), mean absolute percent error (MAPE), accuracy, recall, and precision. Ensemble-learning methods are better when compared to the traditional ML methods, as they can improve model prediction and achieve much better performance [26,27]. It would be beneficial to compare these traditional methods with ensemble methods to investigate potential gains in performance. This study will compare ensemble methods with popular traditional ML methods. The ensemble methods are considered to enhance the performance of the different models.

### 2.2. The Implication of Road Traffic Congestions for Road Traffic Collisions

This section of the study explores the implications of road traffic congestion for road traffic collisions. Road traffic congestion refers to the high number of vehicles trying to use a certain road that cannot handle the number of vehicles [28]. Road traffic congestion mainly happens during peak hours of the day. A road traffic collision happens when a vehicle on a public road collides with another vehicle or involves hitting a stationary object,

a pedestrian, or an animal. Road traffic collisions, which are commonly referred to as "road traffic accidents", can often result in injuries, or even a fatality [29]. The effects of road traffic congestion globally resulting in collisions imposes a burden on the economy and communities. It is well-established in the literature that road traffic congestion and collisions negatively impact road traffic safety [11,30,31]. An increased travel time due to road traffic congestion imposes a high cost to commuters in fuel expedition, replacement of vehicle parts, and much more.

Reducing road traffic congestion and collisions is beneficial to commuters and transport planners, economists, and policy-makers. Researchers [30,31] have studied the relationship between road traffic congestion and collisions; however, there is no consensus regarding the relationship between road traffic congestion and collisions. On the one hand, studies suggest that road traffic congestion is likely to result in light injuries due to the slow vehicular traffic flow. On the other hand, road traffic congestion can lead to high-impact collisions, resulting in serious injuries or fatalities and overall road traffic risks. In summary, it implies that high road traffic congestion is directly propositional to high road traffic collisions resulting in casualties. This can result from various factors, including environmental conditions, the state of the road infrastructure, weather conditions, increased car ownership, stopped vehicles, and also on-road users' behaviour which is out of the scope of the study.

## 3. Methods

This section presents the ensemble methods random forest (RF), decision trees (DT), support vector machine (SVM) and logistic regression (LR) traditional ML methods, and missing data methods considered for this study. Table 1 below contains the RF, DT, SVM, and LR traditional ML methods used as weak learners to be enhanced by the performance of the ensemble methods. Firstly, a classification algorithm is supervised learning utilised to identify the category or grouping of new datasets for the training dataset. In addition, classification is a process used to learn from a given dataset to classify new observations in different classes or groups [32].

**Table 1.** Summary of traditional machine learning (ML) methods.

| No. | ML Method | References |
|:---:|:---:|:---:|
| 1 | Random Forest | [33,34] |
| 2 | Decision Trees | [35,36] |
| 3 | Support Vector Machine | [37,38] |
| 4 | Logistic Regression | [39,40] |

### 3.1. Ensemble Methods

Ensemble methods (EMs) are well-described as ML methods that use multiple base methods and "weak methods" to solve problems. EMs are different from other traditional ML methods in the sense that they use a single dataset to construct more than one hypothesis and then combine them at the end for use [34]. Therefore, the strength of EMs is that they use more than one classifier to enhance the prediction model. EMs are popularly used because they are more robust or accurate than ordinary ML methods, ensuring model robustness. In addition, EMs have been successfully implemented in face recognition, gene expression, and character recognition [41]. This section of the study introduces the ensemble methods together with their characteristics.

#### 3.1.1. Bagging

The bagging ensemble method is a bootstrap that creates its ensemble by creating random samples to train the dataset. Once random samples have been created, the classifier will build a model for each sample. Then, the final results for each sample will be combined

by using what is called majority voting [42,43]. One of the advantages of the method is that it can handle a dataset with missing values.

### 3.1.2. Boosting

The boosting ensemble method mainly focuses on producing a series of methods. The method looks at the training dataset and then chooses its members for the series based on their prediction performance. The method trains the first classifier on the provided dataset; then, the other classifier is trained using the first classifier's surplus. Thus, the method provides a higher weight to the observation that has performed poorly using the previous model. The benefit of the method is that it can achieve more performance rates when compared to the bagging ensemble method [42]. Famous techniques of boosting are adaptive boost, XgBOOST (extreme gradient boosting), and gradient boosting. In this study, the adaptive boosting (AdaBoost) technique will be considered to compute the road traffic congestion models.

### 3.1.3. Stacking

The stacking ensemble method is described as one where predictors, known as level 1 of multiple base classifiers, are implemented as new features for training the meta-classifiers. First, the classifiers are trained, and then their predictions are stacked and used as features for training the meta-classifiers, which will produce the final prediction results [41,44]. A meta-classifier refers to any of the traditional classifiers. In this study, LR has been used as the meta-learner.

### *3.2. Missing Data Method*

In this study, missing values were removed entirely from the real-life dataset using listwise deletion (LD). The LD missing value technique is well-known for completely removing data for the observation that contains one or more missing values [45,46]. The technique applies the assumption that data is missing completely at random (MCAR). LD has been used in this study to produce unbiased estimations even when the dataset is not missing at random. The method is capable of delivering an unbiased estimate.

## 4. Empirical Framework

This section of the study presents the experimental setup, experimental procedure, a summary of the RTC dataset, statistical analysis for the RTC dataset, hyperparameter tuning of the classifiers, and evaluation metrics.

### *4.1. Experimental Settings*

This work aims to develop a predictive model and to compute the misclassification cost of the developed predictive model. In this study, four traditional ML methods, three ensemble methods, one missing-data method, four performance evaluation metrics, and misclassification cost have been used on an RTC real-world dataset. The experiments were conducted using SPyDER from Anaconda Python Distributor. The study aims to combine three ensemble methods with weak RF, DT, SVM, and LR traditional ML methods to observe the different outcomes of the combination in identifying traffic states. Hyperparameter tuning will be encompassed to obtain various results, as presented in Table 2. Figure 1 shows the step-by-step process flow depicting the overall conducted experiments. The main procedures from the experimental process are data preprocessing, statistical analysis, modelling of the different methods, and model performance evaluation.

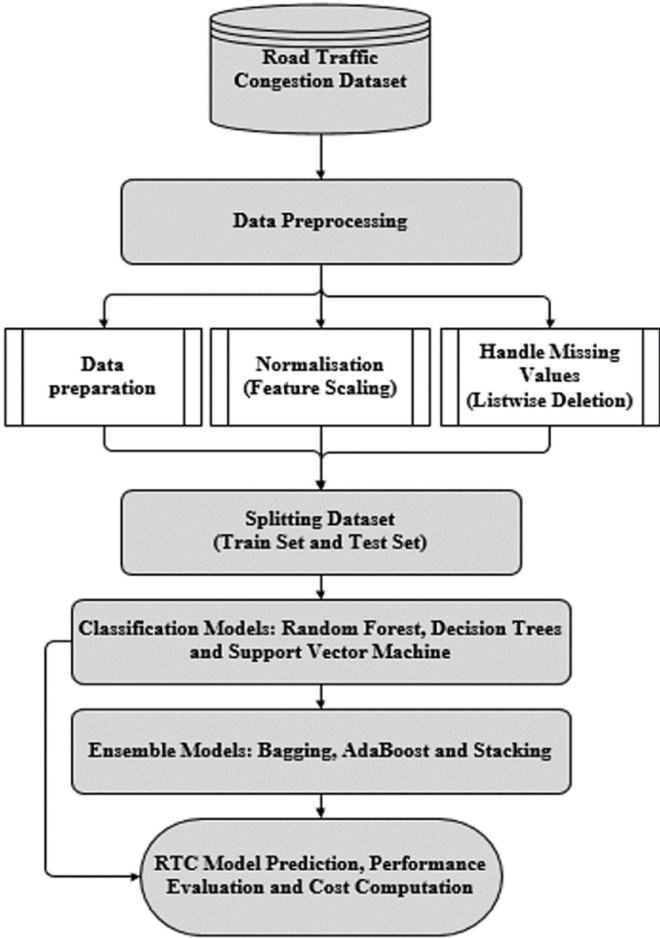

**Figure 1.** Road traffic congestion experimental process.

**Table 2.** Experimental methods for the study.

| No. | Method/Model | Hyperparameter Settings | References |
|---|---|---|---|
| | | Missing Data Method | |
| 1 | Listwise Deletion (LD) | - | [45,46] |
| | | Classifiers | |
| 1 | Random Forest | n_estimators = 100, criterion = 'entropy', random_state = 42, max_depth = 2 | [33,34] |
| 2 | Decision Trees | random_state = 42, max_features = 'auto', criterion = 'entropy', min_samples_split = 100, min_samples_leaf = 1 | [35,36] |
| 3 | Support Vector Machine | kernel = 'poly', degree = 3, C = 1, random_state = 0 | [37,38] |
| 4 | Logistic Regression | Default settings | [39,40] |
| | | Ensemble Models | |
| 1 | Bagging | n_estimators = 1; base_estimator = RF | |
| 2 | Bagging | n_estimators = 1; base_estimator = DT | [42,43] |
| 3 | Bagging | n_estimators = 1; base_estimator = SVM | |
| 4 | AdaBoost | random_state = 0, algorithm = 'SAMME', base_estimator = RF, n_estimators = 1, learning_rate = 5.0 | |
| 5 | AdaBoost | random_state = 0, algorithm = 'SAMME', base_estimator = DT, n_estimators = 1, learning_rate = 5.0 | [47–49] |
| 6 | AdaBoost | random_state = 0, algorithm = 'SAMME', base_estimator = SVM, n_estimators = 1, learning_rate = 5.0 | |
| 7 | Stacking | Final estimator: LR;Classifiers/Estimators: RF, DT & SVM | [41,44,50] |

Different experiment settings are given in Table 2, containing the summary of the different methods and hyperparameter settings. These methods were chosen due to their popularity, characteristics, and frequent citation from the literature.

### 4.2. Road Traffic Congestion Dataset

In this study, the Gauteng Province (GP) road traffic flow dataset was used [51]. The dataset obtained from Mikros Traffic Monitoring (MTM) was used for constructing a vehicle traffic prediction model [52]. MTM is one of the Department of Transport-contracted agencies that collect data directly from the freeways in South Africa. The agency uses data loggers to collect data from the freeways and cameras to monitor traffic volumes on the GP freeways. MTM does not use these datasets to predict road traffic conditions [52]. Approval to conduct the study and collect real-life RTC datasets was attained from MTM and the University of Johannesburg ethics committee (Ethics code: 2020SCiiS04 and Date: 2 March 2020). During data preparation, the collected data was cleaned, normalised, and dealt with missing values and statistical analysis as presented in Figure 1. More so, the traffic flow dataset was categorised into three classes: *Freeflow,* which means that vehicles are travelling at speeds larger than 109 km/h; *moderate* or *Bound flow*, meaning vehicles are entering a bistable traffic state travelling on speeds between 100 km/h and 109 km/h; and *Congested*, meaning travelling at speeds less than 99 km/h. The dataset informed the class categorisation, which lacked the classification of different vehicles involved. The total numbers involved different vehicles that used the specific location. The obtained summary of the dataset is presented in Table 3 and has incorporated features such as *the TravelTime, TrafficVolume, AverageSpeed*, date, and year in Table 4. This paper used data for weekdays only and excluded holiday datasets since they are just simple repeating patterns and are not affecting anyone. Table 3 contains the summary of the overall road traffic congestion dataset.

**Table 3.** Summary of the road traffic congestion dataset.

| Dataset | Instance | Features | Classes | Missing Values | Years |
|---|---|---|---|---|---|
| Road traffic congestion | 50,313 | 5 | 3 | 15,869 | 7 |
| Dataset Split | | | | | |
| Train set (80%) | 27,555 | 5 | 3 | - | 7 |
| Test set (20%) | 6889 | 5 | 3 | - | 7 |

**Table 4.** RTC features and classes descriptions.

| No. | Feature | Description | Data Type |
|---|---|---|---|
| 1 | TravelTime | Refers to the time of day vehicles were travelling | Numeric |
| 2 | AverageSpeed | Refers to the speed at which the vehicles were moving | Numeric |
| 3 | TrafficVolume | Refers to the number of vehicles on the freeway or road | Numeric |
| 4 | Date | Date on which different instances were captured | Date |
| 5 | Year | Different years in which the road traffic dataset was captured | Year |
| | TrafficState | Description | |
| 1 | Congested | Refers to the vehicles being in an unstable traffic state (99 km/h going down) | Ordinal |
| 2 | Moderate or Bound flow | Refers to the vehicles moving towards bistable traffic state (100–109 km/h) | Numeric |
| 3 | Freeflow | Refers to the vehicles moving freely at the maximum required speed (110 km/h going up) | Numeric |

Table 4 contains several features and their descriptions and data type values. These parameters were used to design the RTC predictive model. The dataset contained 50,313 instances with missing values being removed using listwise deletion. The method does not consider missing values compared to other missing data methods dealing with missing values in the dataset. The application of LD to the dataset meant deleting the missing values from the dataset completely, reducing the dataset, which remained with 34,444 instances, and it was split into training and testing sets, as shown in Table 3.

Various studies implemented imputation methods and LD, such as work by [53,54]. These methods can be used to handle "missingness" in the dataset.

RTC Statistical Analysis

Figure 2 shows summary statistics plots for the road traffic congestion dataset, which was performed before removing missing values. Figure 2a shows the three *TrafficStates* known as *Congested*, *Freeflow*, and *Moderate*. It is observed from the bar plot that the *Moderate* class contributes more to the road traffic congestion; this class means that the vehicles are about to move into a *Congestion* state. Figure 2b, the histogram shows how the data is spread out for the *AverageSpeed* feature, with the dataset revealing that most of the dataset is more oriented and skewed between 80 km/h and 120 km/h. This plot means the majority of the dataset is distributed towards the specific *AverageSpeed* range, with most vehicles moving at high speeds between 80 km/h and 120 km/h.

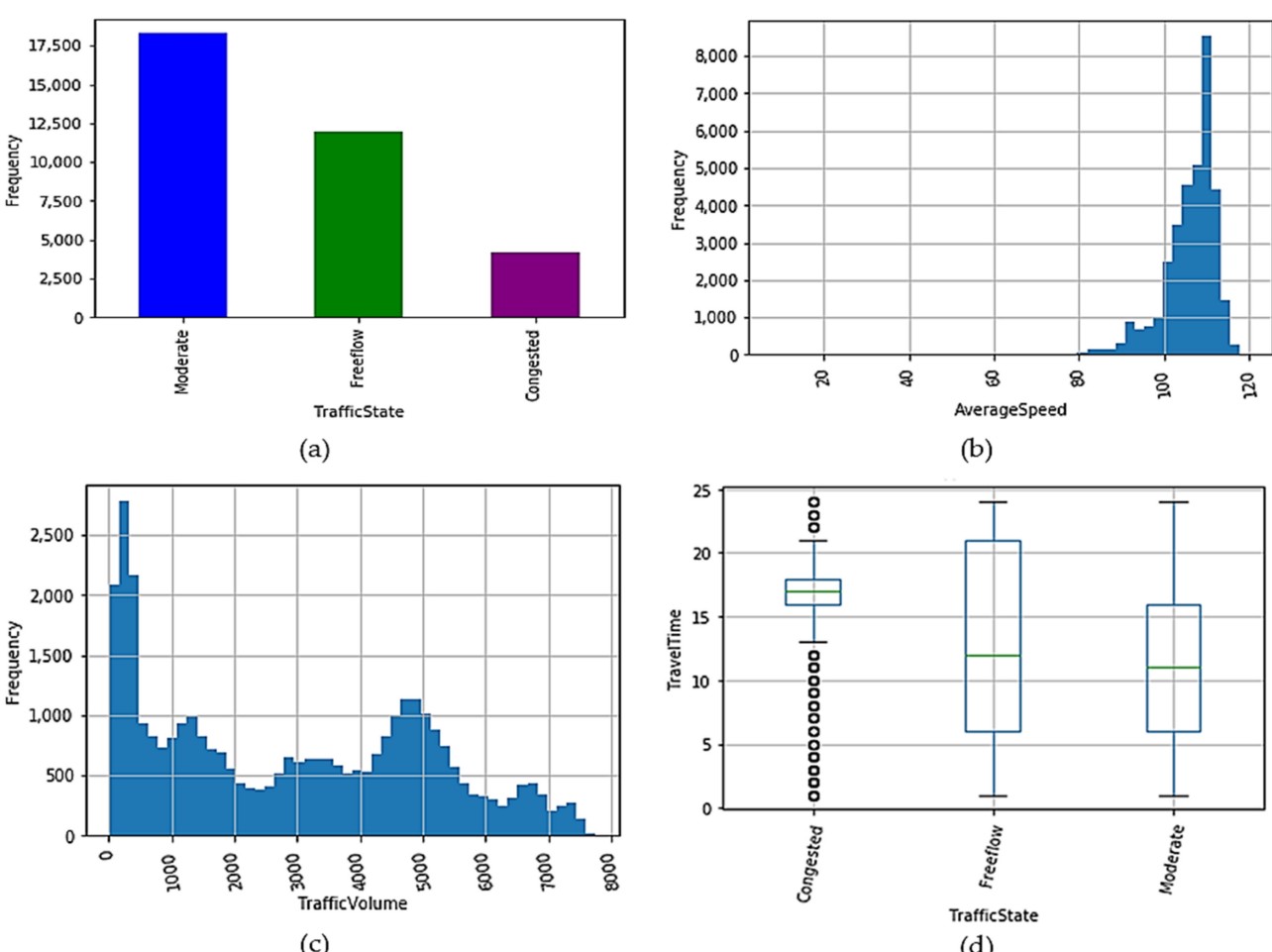

**Figure 2.** Summary analysis plots for RTC dataset. The three *TrafficStates* known as *Congested*, *Freeflow*, and *Moderate* (**a**); *AverageSpeed* (**b**); *TrafficVolume* (**c**); *TrafficState* (**d**).

In Figure 2c, the histogram plot shows *TrafficVolume* frequency skewed to the right, with most of the dataset located on one side. This means that most of the dataset is clustered

on the right side of the plot. From the plot, most of the traffic volume is distributed more towards 5000 and 2000 going down. In Figure 2d, the boxplot shows the *TrafficState* against the *TravelTime* feature; the plot reveals that the *Freeflow* state is distributed throughout the *TravelTimes*. The *Moderate* state is distributed among a few *TravelTimes*, which is less than the *Freeflow* state, and in terms of the *congestion*, the state is distributed between 15:00 and 20:00 *TravelTime* with outliers, which can be a sign of traffic congestion occurring between the specific *TravelTimes*. In general, this means the *Freeflow* traffic state contributes more in terms of the TravelTime feature when compared to the other classes.

Figure 3 shows the total number of instances per *TrafficState* after handling missing values using listwise deletion. Figure 3 shows that *Moderate TrafficState* had 17,675 instances, *Freeflow TrafficState* had 11,962, and *Congested TrafficState* had 4807. With *Moderate* state being the most contributing class, vehicles are moving towards congestion, meaning vehicles are travelling on an open road with no traffic congestion.

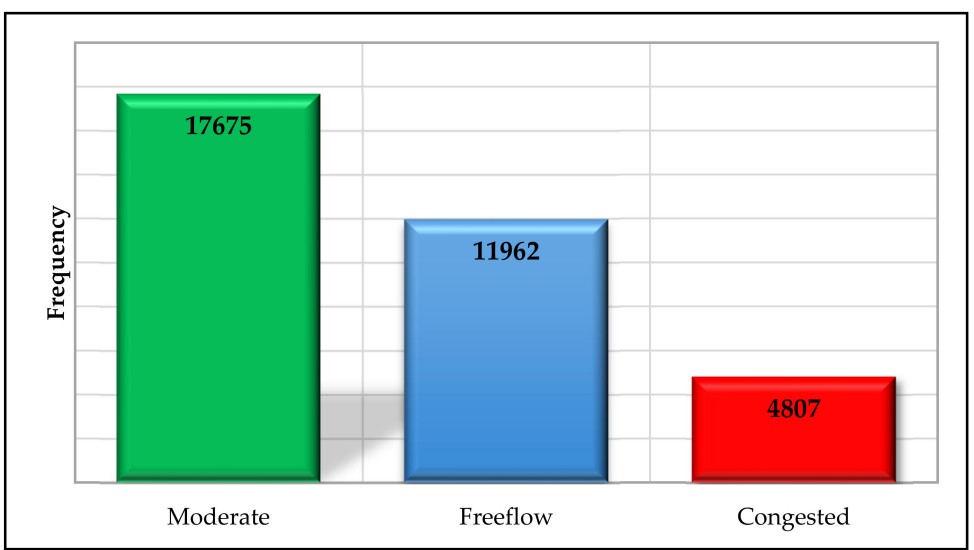

**Figure 3.** RTC totals per TrafficState.

*4.3. Performance Evaluation*

This study used four performance metrics: accuracy, precision, recall, and f1-score. Accuracy is a metric that can be used to evaluate the performance of a classification model. Accuracy is the fraction of the prediction model for the number of correctly predicted and the total number of predicted values [55,56], as shown in Equation (1), which can be computed as follows: TP represents the True Positive, TN represents the True Negative, False Negative (FN), and False Positive (FP).

$$\text{Accuracy} = \frac{\text{TP} + \text{TN}}{\text{TP} + \text{FN} + \text{TN} + \text{FP}} \times 100\% \tag{1}$$

Equation (2) shows the precision measures for the correct positive predictions, divided by all retrieved instances represented by a true positive and false positive. Recall is the measure of how many true positives are correctly predicted out of the entire positives in the RTC dataset, computed using Equation (3). A high value of recall means that most are true positives, and a lower value means that there is an increased number of FN [57]:

$$\text{Precision} = \frac{\text{TP}}{\text{TP} + \text{FP}}, \tag{2}$$

$$\text{Recall} = \text{sensitivity} = \frac{\text{TP}}{\text{TP} + \text{FN}}. \tag{3}$$

F1-score metrics consider both precision and recall and are well-known as the weighted average of both recall and precision metrics, depending on their weight function. It is computed using the following Equation (4):

$$\text{F1} = 2 \times \frac{\text{Precision} \times \text{Recall}}{\text{Precision} + \text{Recall}} . \tag{4}$$

## 5. Experimental Results and Discussion

This section of the study outlines the results obtained during experiments, discussion of the results, and computation of the misclassification cost. Models for predicting the status of vehicle traffic on the freeway were constructed using traditional ML methods and ensemble methods (EMs). The data contained attributes such as travel time, average speed, traffic volume, and date.

### 5.1. Comparison of RTC Prediction Results

Table 5 and Figure 4 show overall results obtained when traditional methods and the bagging method (using RF, DT, and SVM), the AdaBoost method (using RF, DT, and SVM), and the stacking method with a combination of RF, DT, and SVM, with the final estimator as default logistic regression, were used. Results computed by using traditional methods revealed that the DT model obtained more promising results than RF and SVM. Thus, the model performed well in terms of all evaluation metrics. Decision trees proved to have performed well since they could handle datasets with outliers.

**Table 5.** Summary of overall results.

| Ensemble Model | | Metrics | | | |
| --- | --- | --- | --- | --- | --- |
| | | Accuracy | Precision | Recall | f1-Score |
| Model 1 | RF | 0.961 | 0.981 | 0.912 | 0.941 |
| Model 2 | DT | 0.983 | 0.992 | 0.983 | 0.992 |
| Model 3 | SVM | 0.972 | 0.981 | 0.972 | 0.971 |
| Model 4 | Bagging (RF) | 0.963 | 0.972 | 0.912 | 0.931 |
| Model 5 | Bagging (DT) | 0.982 | 0.972 | 0.962 | 0.963 |
| Model 6 | Bagging (SVM) | 0.971 | 0.980 | 0.972 | 0.983 |
| Model 7 | AdaBoost (RF) | 0.961 | 0.980 | 0.913 | 0.942 |
| Model 8 | AdaBoost (DT) | 0.997 | 0.960 | 0.980 | 0.990 |
| Model 9 | AdaBoost (SVM) | 0.521 | 0.510 | 0.352 | 0.262 |
| Model 10 | LR [Stacking (DT, RF and SVM)] | 0.997 | 0.990 | 0.982 | 0.982 |

More results were generated using ensemble methods. The bagging model with DT achieved the best results in terms of accuracy and precision performance metrics. The AdaBoost model with DT also achieved better results than the combination of RF and SVM, which means the combination of AdaBoost with DT significantly improved the model's performance. Additionally, the stacking model, with base estimators as RF, DT, and SVM and the final estimator defined as LR, has shown more significant improvement. In terms of performance, considering accuracy, precision, recall, and f1-score, the model best performed compared with the other models. The results revealed that DT could improve the performance of different ensemble model combinations when used individually and as a weak learner.

A typical analysis example of the state of traffic involving the three vectors/classes is as follows: in 2016, the volume of vehicles travelling on the freeway was recorded as 4987 on a Monday from 06:00, travelling at 87 km/h, resulting in an unstable traffic state.

The unstable traffic state is due to different factors such as commuters travelling to work, road incidents, and other factors. Then, in 2018, the volume of vehicles on the freeway was recorded as 942 travelling at 108 km/h on a Friday from 23:00, which is classified as a bistable traffic state. Furthermore, in 2016, the volume of vehicles on the freeway was recorded as 326, on a Thursday from 05:00, travelling at 110 km/h, classified as stable traffic. During peak hours, traffic turns unstable since most commuters need to get to their different workplaces. Then, during working weekdays in the morning, traffic turns unstable from 06:00 to 08:00 and in the afternoon from 15:00 to 18:00. The analysis shows various peak and off-peak hours to enable commuters to plan for future travelling.

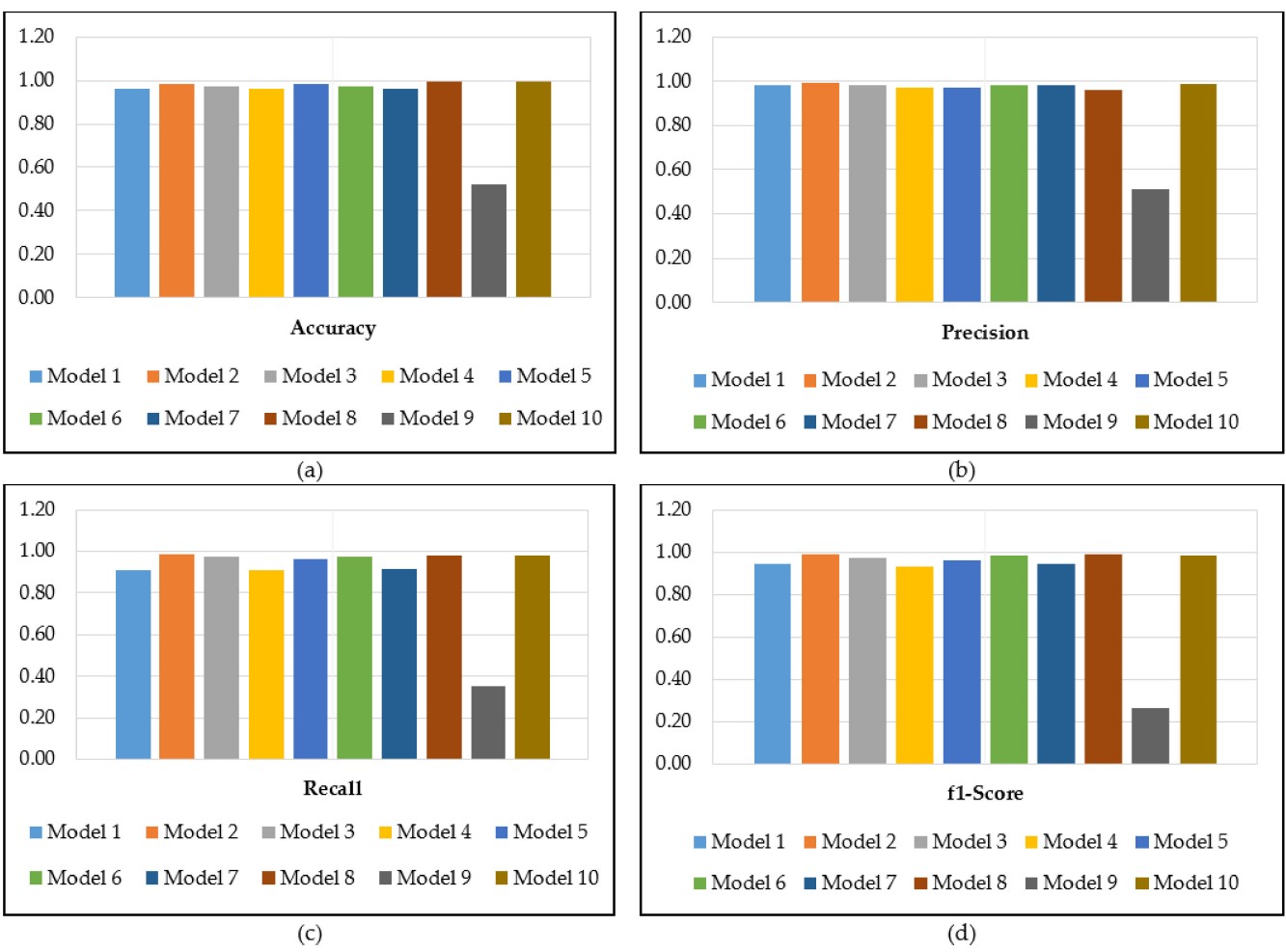

**Figure 4.** Results of the performance metrics (**a**) accuracy, (**b**) precision, (**c**) recall, and (**d**) f1-score for the different models. From the x-axis: Model 1—RF, Model 2—DT, Model 3—SVM, Model 4—Bagging (RF), Model 5—Bagging (DT), Model 6—Bagging (SVM), Model 7—AdaBoost (RF), Model 8—AdaBoost (DT), Model 9—AdaBoost (SVM), Model 10—Stacking (LR-[RF, DT and SVM]).

*5.2. Models' Misclassification Cost*

Misclassification cost refers to all the penalties associated with errors during the classification process. The study aimed to determine the model that did not penalise commuters heavily in travel time and had sufficient information regarding traffic flow. All models were evaluated using the loss matrix shown in Table 6. The penalty was assigned based on the misclassification error for each model [55]. A penalty of 3 was assigned to a cell that predicted *Freeflow* (c) when the actual traffic status was *Moderate or Bound flow* (b) and *Congested* (a). A penalty of 1 was assigned to a cell that predicted *Congested* when the actual traffic status was *Moderate*. Then, a penalty of 0 was assigned to entries on the leading diagonal of the loss matrix, as the actual and the predicted vehicle status were the

same. Since there was a $3 \times 3$ confusion matrix from the three classes (*Freeflow*, *Moderate* or *Bound flow*, and *Congested*), a $3 \times 3$ loss matrix was designed.

**Table 6.** Loss matrix for traffic congestion prediction.

| | | Predicted | | |
|---|---|---|---|---|
| | | a | b | c |
| Actual | a | 0 | 1 | 2 |
| | b | 3 | 0 | 1 |
| | c | 3 | 1 | 0 |

The input in cells ($L_{kj}$) of Tables 7 and 8 specifies the penalty associated with the prediction of class Cj when in fact, it is $C_k$ [55]. For all instances, X which belongs to $C_k$, the expected loss is given by Equation (5) below. $L_{kj}$ is the penalty associated with misclassification for a predicted model.

$$R_k = \sum_{j=1}^{c} L_{kj} \int_{R}' P(x|C_k)\, dx \tag{5}$$

$$R = \sum_{k=1}^{c} L_{kj} P(x|C_k) \tag{6}$$

**Table 7.** The AdaBoost (DT) confusion matrix with a low misclassification rate.

| | | Predicted | | |
|---|---|---|---|---|
| | | a | b | c |
| Actual | a | 929 | 8 | 1 |
| | b | 2 | 3554 | 1 |
| | c | 0 | 7 | 2387 |

**Table 8.** The AdaBoost (SVM) confusion matrix with a high misclassification rate.

| | | Predicted | | |
|---|---|---|---|---|
| | | a | b | c |
| Actual | a | 42 | 896 | 0 |
| | b | 0 | 3557 | 0 |
| | c | 0 | 2394 | 0 |

The actual risk Equation (6) minimises each point x; i.e., when regions are chosen [55,58]. Values of the loss matrix (for cells in Table 6) were chosen by hand, based on views of the knowledge expected in MTM. The misclassification cost of prediction was computed using Equation (7):

$$C_{\cos t\_MLP} = \text{Conf}_{kj} x L_{kj}. \tag{7}$$

$$\text{Cost} = (0 \times 929) + (1 \times 8) + (2 \times 1) + (3 \times 2) + (0 \times 3554) + (1 \times 1) + (3 \times 0) + (1 \times 7) + (0 \times 2387) = 24 \tag{8}$$

$$\text{Cost} = (0 \times 42) + (1 \times 896) + (2 \times 0) + (3 \times 0) + (0 \times 3557) + (1 \times 0) + \\ (3 \times 0) + (1 \times 2394) + (0 \times 0) = 896 + 2394 = 3290 \tag{9}$$

The results were obtained when multiplying the values in Table 6 from the loss matrix with the corresponding cell values of a confusion matrix in Tables 7 and 8. The other eight prediction models in Table 5 were computed using their corresponding confusion

matrix. Table 7 showed that 929, 3554 and 2387 were classified correctly, with 19 incorrect classifications from the model development. Table 8 showed 42 and 3557 correctly classified, with 3290 incorrectly classified. The results of the cost computation showed that the AdaBoost (DT) ensemble model obtained the lowest cost of misclassification of 24 when compared to the AdaBoost (SVM) ensemble model, which obtained the highest cost of misclassification of 3290, as shown in Table 9 and Figure 5. This was computed using Equations (8) and (9) above.

**Table 9.** Summary of cost results.

| Ensemble Model | | Cost |
|---|---|---|
| Model 1 | RF | 252 |
| Model 2 | DT | 98 |
| Model 3 | SVM | 180 |
| Model 4 | Bagging (RF) | 323 |
| Model 5 | Bagging (DT) | 313 |
| Model 6 | Bagging (SVM) | 190 |
| Model 7 | AdaBoost (RF) | 252 |
| Model 8 | AdaBoost (DT) | 24 |
| Model 9 | AdaBoost (SVM) | 3290 |
| Model 10 | LR [Stacking (RF, DT and SVM)] | 43 |

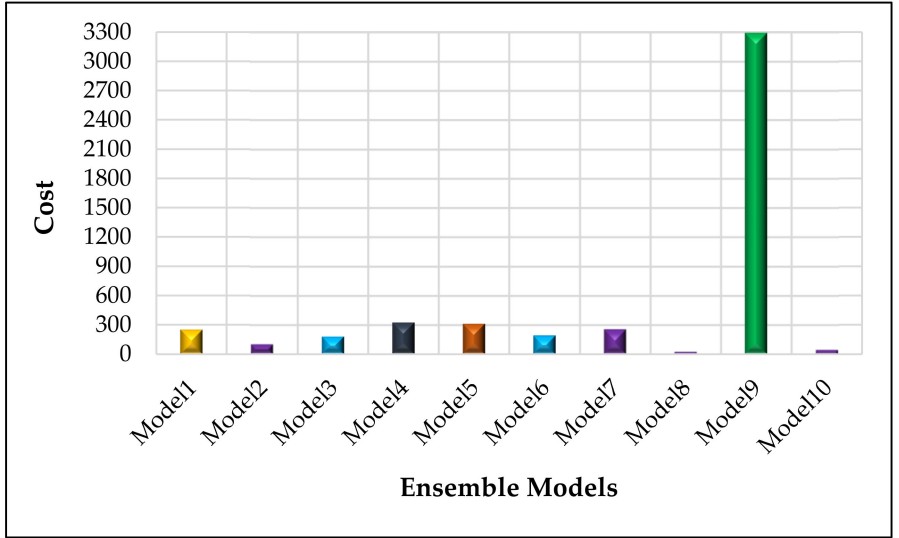

**Figure 5.** The model cost results: cost of RF, DT, SVM, bagging, AdaBoosting, and stacking as shown in the x-axis: Model 1—RF, Model 2—DT, Model 3—SVM, Model 4—Bagging (RF), Model 5—Bagging (DT), Model 6—Bagging (SVM), Model 7—AdaBoosting (RF), Model 8—AdaBoosting (DT), Model 9—AdaBoosting (SVM), Model 10—Stacking (LR-[RF, DT and SVM]).

A road traffic congestion prediction model was developed using ensemble methods. The study results showed that the AdaBoost (DT) ensemble prediction model achieved an accuracy of 99.7% and a prediction misclassification cost of 24, which is lower when compared to the other models. The model obtained the best results in terms of precision, recall, and f1-score metrics. It is hoped that the constructed model would reduce the high number of road traffic collisions and traffic congestion with other interventions already in place. AdaBoost proved to handle the dataset; thus, its performance was as good as expected. This confirmed the suitability of AdaBoost for this kind of dataset and problem

domain. However, the dataset used did not include weather and road collision data. This could be why models performed poorly compared to AdaBoost (DT).

## 6. Conclusions

This paper addressed the determination of vehicle traffic flow well in advance using a dataset from a highway in the Gauteng Province, South Africa. Variables such as travel time, traffic volume, and average speed contributed to a good description of road traffic state regarding its prediction. These were useful for predicting the status of vehicle traffic flow. Results also suggested that the AdaBoost (DT) ensemble model performed better when compared to RF, DT, SVM, bagging, other AdaBoost combination models, and stacking ensemble learning.

This model will complement the current interventions by transport authorities already in place, including e-toll gates, cycle tracks, bus rapid transit, road expansion, and corridors of freedom (well-planned transport routes), among other interventions. Adopting this model will benefit commuters and businesses in the province and potentially make the province an attractive destination for investors. This method is better than the available methods, which have been used to date for addressing these challenges in the Gauteng Province. Although the results here are specific to the dataset used and cannot be generalised, they afford the validation of the presented framework, which may be applied on different road traffic datasets. The predictive model, when implemented well, could directly and indirectly decrease the number of road traffic collisions and risks of poor health in different communities. Furthermore, commuters wishing to travel on the highway will receive helpful information on the traffic flow state well in advance, thanks to the developed prediction model. The proposed prediction model will enable commuters to plan travel time properly. The model can also benefit the authorities to plan the distribution of resources. Businesses will also see improvements due to the timely delivery of goods and staff reporting to work on time, positively impacting the economy of different cities.

For future work, authors are planning to pay more attention to deep learning and various missing data imputation methods as promising emerging approaches, and the authors see this as a future area worth exploring.

**Author Contributions:** T.B.: conceptualisation, data curation, formal analysis, methodology, experiments, writing, editing and results validation. W.D. and B.S.P.: project administration, supervision, conceptualisation and critical revision. All authors have read and agreed to the published version of the manuscript.

**Funding:** This research did not receive external funding.

**Institutional Review Board Statement:** Not applicable.

**Informed Consent Statement:** Informed consent was obtained from all subjects involved in the study.

**Data Availability Statement:** The data are not publicly available due to restrictions from the subjects' agreement.

**Acknowledgments:** The authors would like to thank the Department of Applied Information Systems, the Institute for Intelligent Systems, and the University of Johannesburg for the resources used. The authors also wish to thank Mikros Traffic Monitoring (MTM) (Pty) Ltd. for providing the road traffic dataset used during experiments, and we would also like to thank Eustace Dogo and Barnabas Gatsheni for their useful comments.

**Conflicts of Interest:** The authors do not have any conflict of interest with other entities or researchers.

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
