# Peer review of "A Comparative Study of Ensemble Models for Predicting Road Traffic Congestion"

_applsci, doi:10.3390/app12031337_

Round 1

Reviewer 1 Report

In this paper, the authors studied road traffic congestion prediction using traditional ensemble models.

Comparative experiments using traditional machine learning models and ensemble models are the core contributions of this paper. This paper did not propose a new model, and comparative experiments are all.

Various studies [16][17] such as deep learning methods were conducted on road traffic congestion prediction. However, in this paper, it would be better to include the deep learning model as an experimental comparison target.

Figure 1 is not suitable for publication because of its low resolution.

Overall, it would be nice to increase readability of English expressions easily.

Author Response

Reviewer#1, Concern # 1: Comparative experiments using traditional machine learning models and ensemble models are the core contributions of this paper. This paper did not propose a new model, and comparative experiments are all.

Author response:  Thank you to Reviewer#1 for the comment. Authors agree that the methodology proposed is novel, a comprehensive method of RTC modelling by considering various imputation methods, costs computation, EDA, and performance metrics.

Author action: We updated the manuscript by clearly highlighting the scientific contribution of the study in section 1, paragraph 5.

Reviewer#1, Concern # 2: Various studies [16][17] such as deep learning methods were conducted on road traffic congestion prediction. However, in this paper, it would be better to include the deep learning model as an experimental comparison target.

Author response:  Authors would like to thank the reviewer for the comment. The main focus of the study was to apply traditional machine learning and ensemble methods. We opted for ensemble methods because Deep learning is better suited for a much larger dataset.

Author action: The authors have revised the manuscript and highlighted in the conclusion section 6 that deep learning can be considered for future research with much larger data than the current.

Reviewer#1, Concern # 3: Figure 1 is not suitable for publication because of its low resolution.

Author response:  The authors thank the reviewer for this comment.  

Author action: We updated the manuscript making sure Figure 1 is readable.

Reviewer#1, Concern # 4: Overall, it would be nice to increase readability of English expressions easily.

Author response:  Authors would like to thank Reviewer#1 for this comment.  

Author action: Authors have submitted the manuscript to the language editor for grammar editing.

Reviewer 2 Report

This paper tackles the comparison of methods for traffic congestion prediction. A very up-to-date topic regarding the domain of Intelligent Transportation Systems. Real-world traffic data are used, giving higher merit to this study. However, when dealing with some data sets, you should also emphasize domain-specific knowledge. In your case, you are dealing with traffic flow and traffic congestion. Explanation about the references for traffic congestion detection and characteristic traffic flow behavior is missing. Basics about the fundamental diagram would be welcome. The fundamental explanation is available at https://en.wikipedia.org/wiki/Fundamental_diagram_of_traffic_flow for a starter. Thus, to improve the paper emphasize the following:

  • The abstract should concisely mention the existing problem, how you addressed it and what benefits were achieved. Currently, this is not visible from your abstract. It does not read well in the context of a self-explainable text. Your title emphasizes a comparison of traffic congestion prediction methods. Your abstract does not, and neither is the need to compare such methods explained!
  • It seems you copied a lot of phrases and sentence parts from existing papers, which should be corrected to reduce the similarity to currently published papers, especially in a paper intended for a highly cited indexed scientific journal.
  • Explain each abbreviation when used first. An abbreviation that is common for you is maybe not common knowledge to your reader!
  • Your introduction should contain a motivation description for starting your work and a precise enumeration of your scientific contributions compared to existing knowledge or approaches.
  • Back-up important claims for your research with existing literature!
  • Be precise and concise in writing. Avoid phrases like many, some, … Give particular examples.
  • Add all of the needed details in the experimental settings. For example, how the hyperparameter tuning is done.
  • You state that the speed of 99 km/h or lower is equal to a congested road? This should be elaborated more deeply since in some countries, trucks and busses are limited to 80 or 90 km/h reducing the average speed from the maximal 130 km/h limit for cars. Thus, with more trucks and busses, the average motorway speed will be lower than 100 km/h without the traffic flow entering congestion, especially if you mention that speed above 80 km/h is high!
  • It seems that your data set does not contain much data indicating congestion problems. The lowest measured speed is around 80 km/h, indicating more dense traffic than congested traffic. Especially if the traffic flow also contains trucks and busses.
  • Your results section gives only statistical parameters, and no analysis is done regarding how a typical working day traffic flow looks like, what day parts are congested, and some examples of exactly predicted or classified congestion.
  • Your conclusion mentions that you have improved the prediction result on your local data set. But the main question is if your finding can be generalized, and can your method generally produce a better output compared to other methods?

I recommend you make thorough proofreading using an English native speaker as help to improve the clarity and consistency of your written text.

There are additional comments in the attached PDF.

Author Response

Reviewer#2, Concern # 1: However, when dealing with some data sets, you should also emphasize domain-specific knowledge. In your case, you are dealing with traffic flow and traffic congestion. Explanation about the references for traffic congestion detection and characteristic traffic flow behavior is missing. Basics about the fundamental diagram would be welcome. The fundamental explanation is available at https://en.wikipedia.org/wiki/Fundamental_diagram_of_traffic_flow for a starter.

Author response:  Thank you to Reviewer#1 for the comment. The context of the scientific domain has now been enhanced, and information about traffic congestion has been added to section 1 with extra references.

Author action: We updated the manuscript by adding paragraphs 3 and 5 in Section 1.

Reviewer#2, Concern # 2: The abstract should concisely mention the existing problem, how you addressed it and what benefits were achieved. Currently, this is not visible from your abstract. It does not read well in the context of a self-explainable text. Your title emphasizes a comparison of traffic congestion prediction methods. Your abstract does not, and neither is the need to compare such methods explained!

Author response:  Authors would like to thank the reviewer for the comment.

Author action: We updated the manuscript by revising the abstract to highlight the problem and benefits of the study to correspond with the title.

Reviewer#2, Concern # 3: It seems you copied a lot of phrases and sentence parts from existing papers, which should be corrected to reduce the similarity to currently published papers, especially in a paper intended for a highly cited indexed scientific journal.

Author response:  The authors thank the reviewer for this comment.  

Author action: We updated the manuscript by revising Sub-Section 2.1 related studies.

Reviewer#2, Concern # 4: Explain each abbreviation when used first. An abbreviation that is common for you is maybe not common knowledge to your reader!

Author response:  Authors would like to thank Reviewer#1 for this comment.  

Author action: We updated the manuscript by ensuring each abbreviation has been explained the first time of use.

Reviewer#2, Concern # 5: Your introduction should contain a motivation description for starting your work and a precise enumeration of your scientific contributions compared to existing knowledge or approaches.

Back-up important claims for your research with existing literature!

 Author response:  Authors would like to thank the reviewer for the comment.

Author action: We have updated the manuscript by adding the contribution paragraph 5 in Section 1.

Reviewer#2, Concern # 6: Be precise and concise in writing. Avoid phrases like many, some, … Give particular examples.

Author response:  Authors would like to thank the reviewer for the comment.

Author action: We updated the manuscript by making sure such phrases were removed.

Reviewer#2, Concern # 7 Add all of the needed details in the experimental settings. For example, how the hyperparameter tuning is done.

Author response:  Authors would like to thank the reviewer for the comment.

Author action: We update the manuscript by clarifying that Table 2 contains all the hyperparameter tuned parameters for the methods.

Reviewer#2, Concern # 8: You state that the speed of 99 km/h or lower is equal to a congested road? This should be elaborated more deeply since in some countries, trucks and busses are limited to 80 or 90 km/h reducing the average speed from the maximal 130 km/h limit for cars. Thus, with more trucks and busses, the average motorway speed will be lower than 100 km/h without the traffic flow entering congestion, especially if you mention that speed above 80 km/h is high!

It seems that your data set does not contain much data indicating congestion problems. The lowest measured speed is around 80 km/h, indicating more dense traffic than congested traffic. Especially if the traffic flow also contains trucks and busses

Author response:  Authors would like to thank Reviewer#2 for the comment. The statement we made here is specifical regards the data, which can vary from country to country.

Author action: We updated the manuscript by revising sub-section 4.2 by clarifying reasons why specific km/h were defined for this study.

Reviewer#2, Concern # 9: Your results section gives only statistical parameters, and no analysis is done regarding how a typical working day traffic flow looks like, what day parts are congested, and some examples of exactly predicted or classified congestion.

Author response:  Authors would like to thank Reviewer#2 for the comment.

Author action: We updated the manuscript in section 5.1 by adding the statistical analysis of the parameters.

Reviewer#2, Concern # 10: Your conclusion mentions that you have improved the prediction result on your local data set. But the main question is if your finding can be generalized, and can your method generally produce a better output compared to other methods?

Author response:  Authors would like to thank Reviewer#2 for the comment. The results are clearly linked to the dataset but cannot be generalized. The obtained results depend on the type of data, ML method and hyperparameter tuning.

Author action: We updated the manuscript by clarifying this in Section 6.

Reviewer#2, Concern # 11: I recommend you make thorough proofreading using an English native speaker as help to improve the clarity and consistency of your written text.

Author response:  Authors would like to thank Reviewer#2 for the comment.

Author action: Authors have sent in the manuscript for language editing.

Reviewer#2, Concern # 12: What about the scientific contribution of your study?

Author response:  Authors would like to thank Reviewer#2 for the comment.

Author action: We updated the manuscript by editing Section 1 with paragraph 5.

Reviewer#2, PDF comments:

  • Add corresponding author mark
  • Usually, if authors are from the same institution notation I used. Thus, use only 1 and 2 for institution notations
  • Unclear sentence (Abstract)
  • A traffic prediction model can do this only if used with other approaches like traffic control, vehicle routing….(Abstract)
  • Explain each abbreviation on when used first.
  • Add here a paragraph describing the motivation for using road traffic congestion prediction and how your reseach fits into that (Introduction)
  • What about the scientific contribution of the study? (Introduction)
  • You must back-up such statements with literature! (Section 2.2)
  • Be precise, mention the methods you use (Section 3)
  • What do you mean by stronger? More robust to bad input data, more accurate, easier to learn. (Section 3.1)
  • In your paper or in the cited references? (Section 3.1.3)
  • Use the abbreviation when you have defined it (Section 3.3).
  • Mention here which methods will be used for this (Section 4.1)
  • How do you differentiate between moderate and congested? (Section 4.2)
  • Why? Not enough data, no interesting traffic behavior or just simple repeating patterns outside working days? (Section 4.2)
  • The above table mentions congested state with velocity below 99km/h and here you state tat speed above 80km/h is high? (Section 4.2.1)
  • It is better to organize the x-axis from 2013 to 2019 increasingly (Section 4.2.1)
  • Unclear sentence (Section 4.3).
  • Vehicle contribute to road traffic! The variables you mention contribute to a good description of road traffic state or it prediction (Section 6).
  • Ok, on local data your method is better. What about other datasets? Can your finding be generalized? (Section 6)

Author response:  Authors would like to thank Reviewer#2 for the comment. We went through all the comments in the PDF as suggested by reviewer #2.

Author action: We updated the manuscript by clarifying specific sentences, using relevant abbreviations accordingly in Sections such as the Abstract, 2.2,3,3.1, 3.3, 3.1.3,4.1,4.2,4.2.1, 4.3, 6. We also updated the manuscript in section 1 by adding an extra paragraph suggested by reviewer #2 discussing how this study fits in road traffic congestion prediction.

Round 2

Reviewer 1 Report

The authors tried to apply the review results to the paper. 

Author Response

Reviewer#1, Concern # 1: The authors tried to apply the review results to the paper. 

Author response:  Thank you to Reviewer#1 for the comments.

Author action: No action was taken.

Reviewer 2 Report

The authors have improved this paper. However, there are still issues to be corrected. Consider the following:

- Consider that some authors use stable, bi-stable, and unstable traffic flow notations when using the fundamental diagram. Thus, you introduce something that already exists. Consult the paper 

https://doi.org/10.1287/opre.51.5.681.1675 for details. The corresponding paragraph in the Introduction should be corrected.

- There are some abbreviations left undefined.

- The added paragraph in the discussion section is hard to understand regarding the distribution of vehicle speeds. Also, details about the found specific hours should be given and elaborated considering possible driver behavior change regarding the planning of their trips.

Your prediction model can improve the traffic situation only if mobility users apply it correctly to change their trip times and avoid congested routes. Your conclusion should emphasize this, especially since your paper does not consider improving the traffic situation and deals only with prediction. It would be best to concentrate on that part, wrapping up which approach is the best, why, and what shortcomings should be researched in the future.

There are additional comments in the attached PDF.

Author Response

Reviewer#2, Concern # 1: Consider that some authors use stable, bi-stable, and unstable traffic flow notations when using the fundamental diagram. Thus, you introduce something that already exists. Consult the paper https://doi.org/10.1287/opre.51.5.681.1675 for details. The corresponding paragraph in the Introduction should be corrected

Author response:  Thank you to Reviewer#2 for the comment. We could not open the URL given to us by reviewer #2, so we used related papers.

Author action: We updated the manuscript by revising the paragraph in the introduction section. We clearly explained how the study utilized the three vectors of the fundamental diagram of traffic flow.

Reviewer#2, Concern # 2: There are some abbreviations left undefined.

Author response:  Authors would like to thank the reviewer for the comment.

Author action: We updated the manuscript by defining all the abbreviations.

Reviewer#2, Concern # 3: The added paragraph in the discussion section is hard to understand regarding the distribution of vehicle speeds. Also, details about the found specific hours should be given and elaborated considering possible driver behavior change regarding the planning of their trips. Your prediction model can improve the traffic situation only if mobility users apply it correctly to change their trip times and avoid congested routes.

Author response:  The authors thank the reviewer for this comment.  

Author action: We updated the manuscript by revising the conclusion section paragraph and the

Reviewer#2, Concern # 4: Your conclusion should emphasize this, especially since your paper does not consider improving the traffic situation and deals only with prediction. It would be best to concentrate on that part, wrapping up which approach is the best, why, and what shortcomings should be researched in the future.

Author response:  The authors thank the reviewer for this comment. 

Author action: We updated the manuscript by revising the conclusion section.

Reviewer#2, PDF comments:

  • All Abbreviations should be explained in the text when used first. Not all readers are familiar with abbreviations/methods from the ML domain. (Section 3)
  • Unclear part of the sentence (Section 5)
  • Give more details about the found specific hours and corresponding time slots! (Section 5)
  • This paragraph is hard to understand regarding the number of vehicles and traveling speeds. (Section 5)
  • This is very common knowledge and should not be mentioned in a conclusion. (Section 6)
  • How does it do that? By predicting congestion? The application of the model by the users can do this if users decide to travel outside congested intervals. (Section 6)

Author response:  Authors would like to thank Reviewer#2 for the comment.

Author action: We updated the manuscript by revising all the comments as suggested by reviwer #2 in the PDF.

Round 3

Reviewer 2 Report

The paper is improved. Before sending the final version for publishing, change all hour notations to “6:00”. At some places, the notation “6h00” has been used.